# Biosorbents Based on Biopolymers from Natural Sources and Food Waste to Retain the Methylene Blue Dye from the Aqueous Medium

**DOI:** 10.3390/polym14132728

**Published:** 2022-07-03

**Authors:** Alexandra Cristina Blaga, Alexandra Maria Tanasă, Ramona Cimpoesu, Ramona-Elena Tataru-Farmus, Daniela Suteu

**Affiliations:** 1Department of Organic, Biochemical and Food Engineering, “Cristofor Simionescu” Faculty of Chemical Engineering and Environment Protection, “Gheorghe Asachi” Technical University of Iasi, Prof. Dr. docent D. Mangeron Blvd., No. 73, 700050 Iasi, Romania; acblaga@tuiasi.ro (A.C.B.); alexandra_tanasa20@yahoo.com (A.M.T.); 2Department of Materials Science, Faculty of Materials Science and Engineering, “Gheorghe Asachi” Technical University of Iasi, Prof. Dr. docent D. Mangeron Blvd., No. 41, 700259 Iasi, Romania; ramona.cimpoesu@academic.tuiasi.ro; 3Department of Chemical Engineering, “Cristofor Simionescu” Faculty of Chemical Engineering and Environmental Protection, “Gheorghe Asachi” Technical University of Iasi, Prof. Dr. docent D. Mangeron Blvd., No. 73, 700050 Iasi, Romania; ramona-elena.tataru-farmus@academic.tuiasi.ro

**Keywords:** biosorption, methylene blue, immobilization, isotherms models

## Abstract

The use of a biosorbent based on residual biomass from brewing industry (*Saccharomyces pastorianus*) immobilized in a natural biopolymer (sodium alginate) was investigated for Methylene Blue removal from aqueous medium. *Saccharomyces pastorianus*, immobilized by a simple entrapment technique and by microencapsulation in alginate was characterized using SEM, EDAX, pH_PZC_ and the biosorption behavior toward organic pollutant, such as cationic dye. The biosorption experiments were studied by assessing, in a first stage, the influence of the most important operational physical parameters on the efficiency of the biosorbent: the initial concentration of the dye, the contact time between phases, the temperature, the dye solution pH, the biosorbent granule size, and the amount of biosorbent. The highest sorption capacity was obtained for the biosorbent obtained by microencapsulation, at pH 9, at biosorbent dose of 5.28 g/L and a contact time of about 100 min. The biosorption equilibrium was then studied by modeling the data on the Langmuir, Freundlich and Dubinin- Radushkevich isotherms. The Langmuir model is best suited for experimental data on both particle sizes leading to a maximum biosorption capacity of 188.679 mg/g at room temperature. The values of the adsorption energy, E, obtained with the help of the Dubinin-Radushkevich model-suggest that the type of mechanism (physical or chemical) involved in the biosorption process depends on the particle size of the biosorbent. The results confirm that the residual microbial biomass of *Saccharomyces pastorianus* immobilized in a polymeric matrix such as sodium alginate, can be considered an efficient biosorbent in retaining cationic organic dyes present in aqueous solutions in moderate concentrations.

## 1. Introduction

Microbial biosorbents obtained from bacterial, yeast, or filamentous fungi cells [1,2,3,4,5] are used for their ability to bind pollutants (heavy metals, dies, antibiotics) by ionic interactions between the functional groups from the polymers contained in the extracellular surface of the dead (inactive) or living cells and the pollutant, depending on many factors such as pollutant chemical properties and structure, type of biomass (size and structure), pH, temperature, mixing, pollutant concentration, and biosorbent quantity [1]. Microorganisms have the ability to enzymatically degrade dye molecules and have high dye sorption capacity, offering the possibility for complete removal of dye pollutants by association of biosorption and biodegradation [6].

The cell walls of microbial cells (bacteria or fungi) are different in structure, but usually have a multipolymeric structure that fulfils important and diverse functions within it. Bacterial cells possess not only peptidoglycan as a fundamental polymer (responsible for the maintenance of cell shape and osmotic stability, in different proportion in Gram positive and Gram negative bacteria) but also other polymers linked to the peptidoglycan chains: polysaccharides (teichoic or teichuronic acids and other neutral or acidic polysaccharides) and proteins [7]. The fungi cell wall contains a complex mixture of proteins and polysaccharides, including β1,3-glucan (the major component), mannans, and chitin, with roles in maintenance of cell rigidity and shape and metabolism [8].

These polymers have numerous functional groups: carboxyl, phosphate, amine, imidazole, sulfate, glycosyl, sulfhydryl, and hydroxyl groups responsible for pollutant binding through physical adsorption, ion exchange, complex formation, reduction, or precipitation [9].

The use of microbial biomass as biosorbents offers a series of advantages (high sorption capacity, low operating costs, and easiness of use) but is not used on a large scale due to some particular characteristics: small size, low density that arise difficulties in water-biosorbent separation, and insufficient mechanical stability [10]. Inactive (dead) microbial biomass is much easier to use compared to active (living) cells, as it does not require a nutrient medium or maintenance of pure microbial cultures, it can be used over a wide pH or temperature range, the process can be performed in a simple equipment, and since it can be obtained as a byproduct from the biosynthetic industry (production of antibiotics, enzymes, organic acids, amino acids or food products—wine or beer), it involves a low cost.

The challenges of using microbial cells for the production of biosorbents using natural and agricultural resources can be overcome by microbial immobilization using a natural polymer, as alginate or chitosan (biodegradable, biocompatible, economical, and environmentally friendly) for the sorption of various pollutants [11]. Alginate can be used as adsorbent for metal ion removal—Pb (II), Cu (II), Cd (II) [12], but its sorption capacity can be significantly improved by the addition of microbial polymers.

The immobilization process facilitates biosorbent manipulation (increased mechanical stability) and separation at the end of the sorption process, and several techniques can be used [13]. From these, entrapment in polymeric matrices, using alginate by simple immobilization (biomass and carrier suspension dripping in calcium chloride solution with the production of large beads) or microencapsulation (biomass is restricted by a membrane wall in microspheres with the help of a microencapsulator), offers a suitable biosorbent (usually in a bead forms) for technical applications [10,11,14,15,16,17].

*Saccharomyces pastorianus* is an industrial waste from brewery industry (production of lager beers [18]), available in high quantities. Its use as biosorbent has been investigated for dye and pharmaceuticals retention [14,15,16], showing its potential.

Several biosorbents were analyzed for methylene blue retention from effluents: *Lactarius piperatus,* (wild macrofungus) with a biosorption percentage of 86% obtained for T = 25 °C and pH = 7 [19]; *Microspora* sp (algal biomass) with retention of 100% in 24 h [13]; *Chlamydomonas moewusii* (microalga) with a biosorption percentage of 68.5% at pH 10 [20]; *Cunninghamella echinulata UCP* 1297 (filamentous fungus) with 95.4% efficiency in dye removal for pH = 9 and 200 rpm [21].

Methylene blue is a cationic water-soluble dye with multiple uses in the production of textiles, plastic, paper, pharmaceuticals, and cosmetics [18,22,23], but due to its complex structure, natural degradation is difficult (nonbiodegradable); moreover, it is toxic and carcinogenic above certain concentrations [22]. Methylene blue’s presence in effluents (highly visible at small amounts of dyes < 1 ppm), resulted from dying processes or production of pharmaceuticals or cosmetics imposes the use of environmentally friendly, efficient technology for its removal from wastewater. Traditional approaches include [18,24,25]: precipitation, coagulation, photocatalytic degradation, ultra and nanofiltration, or electrochemical treatment. An attractive alternative to these techniques used for treating wastewater is biosorption [18,19,20,21,24,25,26,27,28,29,30,31,32], an adsorption process using biological material for retaining pollutants from water.

In this context, *Saccharomyces pastorianus,* immobilized by a simple entrapment technique and by microencapsulation in alginate was investigated as biosorbent for the retention of methylene blue from aqueous solutions. The biosorbents obtained by immobilization of this industrial waste (residual biomass from brewing industry) was analyzed using SEM, and its efficiency in the biosorption process was experimentally determined.

## 2. Materials and Methods

### 2.1. Materials

The biosorbents used in this research were obtained from industrial biomass waste, *Saccharomyces pastorianus* (Albrau, Onesti, Romania), immobilized using sodium alginate in a simple dripping technique and using a Buchi microencapsulator (Buchi Labortechnik AG, Flawil, Switzerland) following previously detailed procedures [14,16] and systematized in Figure 1. The residual biomass from brewing industry (*Saccharomyces pastorianus*) used in the experiment, was separated from the fermentation broth by centrifugation at 8000 rpm, washed with distilled water, dried at 80 °C (increases the surface area, by cell rupture determining higher binding capacity) and immobilized into sodium alginate, though two methods. One immobilization technique (I) implied simple dripping of a suspension of 5% d.w. residual biomass prepared in 1% sodium alginate solution and dripped after complete homogenization into 1% calcium chloride (prepared in distilled water at 5 °C), through a capillary for the production of spherical beads with F1 = 4 mm diameter. The second method (II) implied the use of a Buchi microencapsulator for the production of beads. In this case, 5% d.w. residual yeast was prepared in 1.5% low viscosity sodium alginate solution and fed to the microencapsulator. The nozzle diameter used was 750 μm, with the following conditions: air pressure 100 mbarri, T = 45 °C, 500 V, and 800 Hz and allowed the production of beads with 1.400 mm diameter. The spherical droplets were collected into a polymerization bath (100 mM calcium chloride solution).

Adsorbate. A cationic dye, methylene blue (MB-Basic Blue 9; 52015) (M_W_ = 319.85 g/mol, λ_max_ = 660 nm, from Merck, KGaA, Darmstadt, Germany) with chemical and molecular structure showed in Figure 2, was selected as chemical pollutant of aqueous system for this study.

The stock solution with a concentration of 500 mg dye/L was prepared from the commercial form of the dye powder. From this, the working solutions were prepared by appropriate dilution with distilled water.

### 2.2. Methods

#### 2.2.1. Batch Biosorption Methodology

Experimental biosorption studies were performed by a simple technique that consists in contacting different amounts of biosorbent with 5% dry matter (d.w.), prewashed with distilled water to remove traces of calcium chloride solution (which could cause precipitation of the dye), with 25 mL of cationic dye solution having different initial concentrations (in the range of 12.8–83.2 mg/L) (Figure 1). The pH values were adjusted with 1N HCl and/or 1N NaOH solution and the set temperature (5 °C, 17 °C, 40 °C) was kept constant using a cooler (for 5 °C) and a thermostatic oven Poleko SLW53 model (Pol-Eko-Aparatura sp.j., Wodzisław Śląski, Poland), for 17 °C and 45 °C. The contact time of the phases was between 8–24 h. The amount of dye present in the solution at the time of equilibrium was determined spectrophotometrically using a Shimadzu UV-1280 UV-VIS spectrophotometer (Shimadzu Corporation, Kyoto, Japan) at a maximum dye wavelength of 660 nm, based on the calibration curve in compliance with the Lambert–Beer law. To evaluate the potential biosorption capacity of the materials studied in the case of laboratory experiments, two characteristic parameters were calculated:the biosorption capacity (q, mg of dye/g of biosorbent):
(1)$$q=\frac{C_{0}-C}{G}\cdot V$$

the percentage of dye removal (R),

R = [(C_0_ − C)/C_0_] × 100 (%)(2)
where C_0_ and C are the initial and the equilibrium (residual) dye concentration in solution (mg/L), G is the amount of biosorbent (dry matter (d.w.) from alginate beads) (g) and V is the solution volume (L). All the biosorption experiments were done in triplicates, the results presented being the average of the obtained values. The errors were between 2.36–5.68%.

#### 2.2.2. Physicochemical Characterization of Biosorbents

The characterization of the biosorbents obtained by the two techniques of immobilization of residual *Saccharomyces pastorianus* biomass in sodium alginate was made to visualize their internal structure, as a result of biomass incorporation in the polymeric structure, using Scanning Electron Microscopy (SEM).

Biosorbent characterization studies based on residual microbial biomass immobilized in sodium alginate, having diameter φ = 4 mm and φ = 1.5 mm, were performed on freeze-dried samples. The lyophilization was performed using a Labconco lyophilizer (Labconco, Kansas City, MO, USA) with the following process parameters: 0.05 mbar; −50 °C, time-6 h.

Scanning electron microscopy (SEM) was carried out to characterize the surface micromorphology of the biosorbent based on *Saccharomyces pastorianus* immobilized in sodium alginate before the biosorption process. A scanning electron microscope, VegaTescan LMH II (Tescan Orsay Holding, Brno—Kohoutovice, Czech Republic), detector SE, WD 15.5 mm, 30 kV, HV, VegaTC software (Tescan Orsay Holding, a.s., Brno, Czech Republic) was used.

#### 2.2.3. The Biosorption Equilibrium Data Analysis

The equilibrium data of the biosorption process of cationic dye methylene blue onto biosorbent obtained by immobilization of residual microbial biomass of *Saccharomyces pastorianus* on alginate sodium matrix were analyzed using three of the most known sorption equilibrium models from scientific literature [33]:➢Freundlich—takes the heterogeneity of the surface and the exponential distribution of the active sites of the biosorbent into account. The general equation is:
(3)$$q=K_{F}\cdot C_{}^{1/n}$$
with the linearized form:(4)$$\log q=\log K_{F}+\frac{1}{n}\log C_{}$$
where: K_F_ and 1/n—are Freundlich constants associated with the biosorption capacity and intensity (efficiency), respectively; a favorable biosorption corresponds to a value of 1 < n < 10.

➢Langmuir—starts from the idea that the maximum biosorption corresponds to a monolayer of solute molecules on the biosorbent surface, which contain a finite number of energetically equivalent sites. The general equation is:

(5)$$q=\frac{K_{L}\cdot C\cdot q_{0}}{1+K_{L}^{}\cdot C}$$
with two linearized forms:

**Langmuir I:** 1/q = f (1/C)
(6)$$\frac{1}{q}=\frac{1}{q_{0}}+\frac{1}{K_{L}\cdot q_{0}}\cdot \frac{1}{C}$$

**Langmuir II:** C/q = f (C)
(7)$$\frac{C}{q}=\frac{1}{q_{0}\cdot K_{L}}+\frac{C}{q_{0}}$$
where: q_0_ and K_L_ are Langmuir constants, q_0_ is the maximum amount of biosorbed solute (mg/g), and K_L_ is the constant related to the binding energy of solute (L/mg)

➢Dubinin–Radushkevich—allows the appreciation of the nature of the biosorption process (physical or chemical) depending on the value of the adsorption energy, E. Thus E < 8 kJ/moll characterizes a physical biosorption mechanism, and values between 8 and 16 kJ/mol suggest an ion exchange mechanism. The general equation is Equation (8).

(8)$$q=q_{0}\exp\left(-B\cdot \varepsilon^{2}\right)$$
with generalized form:
ln q = ln q_0_ − Bε^2^(9)
(10)$$\varepsilon =\mathrm{RT}\;\ln\left(1+\frac{1}{C}\right)$$
(11)$$E=\frac{1}{\sqrt{2B}}$$
where: q_D_ is the maximum biosorption capacity (mg/g); β is the activity coefficient related to mean biosorption energy; ε is the Polanyi potential (ε = RTLn (1 + 1/C)); E is the mean free energy of biosorption (kJ/moll); T (°K) is the absolute temperature; R (J/(moll K) is the gas constant; and C (mg/L) is the concentration of dye in solution at equilibrium.

In order to obtain the quantitative characteristics of each model, the linearized forms of the general equations were graphically represented. From the intersection with the “y” axe and the slope of the obtained line, the values of the characteristic parameters were determined.

## 3. Results

### 3.1. Preparation of Microbial Biosorbent and Their Physical–Chemical Characterization

The SEM images, obtained at 500 and 1000× magnifications and presented in Figure 3, highlight the morphology, and pores distribution. Porosity and large surface area are important elements in the case of materials used as biosorbents for chemical pollutants. The SEM imagines for biosorbents reveal the existence of a mesoporous structure of the studied material. The representations in Figure 3 indicate particles with a regular geometric shape. Both types of beads present compact, rough surface, and high porosity, but the microspheres obtained through microencapsulation possess a higher density, surface roughness being more pronounced compared to the particle obtained by simple dripping, which presents a good prospect for biosorption.

Energy-dispersive X-ray spectroscopy (EDAX EDS, EDX, EDXS, or XEDS technique) technique was used for the elemental analysis and chemical characterization of the biosorbents obtained by immobilization of residual microbial biomass of *Saccharomyces pastorianus* using two techniques: a simple dripping technique and microencapsulation using a Buchi microencapsulator.

The EDAX spectrum obtained for the characterized samples showed the presence on the surface of various elements that come from the structure of polymeric composites based on microbial biomass and sodium alginate (Figure 3a,b). The amounts of calcium are due to the fact that the prepared granules, regardless of the technique, are stored in 2% CaCl_2_ for stabilization until their use in the biosorption process, where they are washed before use with distilled water. During this storage, an ion exchange takes place; respectively, the Ca^2+^ ions have replaced the Na^+^ ions of the sodium alginate for the gelation to take place, so the calcium becomes part of the material. Chloride ions, on the other hand, are present in most alginate samples.

### 3.2. The pH_PZC_ Value

The behavior of the biosorbent against different categories of anionic and/or cationic pollutants is determined by the electric charge of its surface, a point called zero charge point (pH_PZC_). The pH_PZC_ (zero charge pH) value for the biosorbent based on the residual biomass immobilized in sodium alginate was determined using the method proposed by Nouri and Haghseresht and is 5.4 [11]. Thus, for pH values < pH_PZC_, the surface of the material will be positively charged due to the high concentration of H^+^ ions (characteristic groups are positively charged) and will be able to react with anionic species based on electrostatic interactions and hydrogen bonds. In the case of pH > pH_PZC_ values, the surface is negatively charged due to the dissociation of characteristic functional groups and will be able to retain cationic species by ion exchange and/or electrostatic interactions. According to these observations, the optimal retention of the cationic dye is expected to be achieved at a basic pH.

### 3.3. Evaluation of the Biosorbent Potential of the Obtained Microbial Biosorbents

#### Effect of the Main Physical–Chemical Operating Parameters on Biosorption of Methylene Blue Dye onto Microbial Biosorbent

In order to study the biosorption efficiency of the synthesized materials by immobilization of residual biomass in sodium alginate matrix, the influence of some physico-chemical operating parameters on the biosorption process of methylene blue dye was analyzed. The main factors taken into consideration were the pH, the amount of sorbent, the temperature, the phases contact time, and the initial dye concentration in the aqueous solution (Table 1). This study allows to establish the operating conditions in which a higher biosorption capacity is obtained.

The obtained results were systematized in Figure 4.

The following conclusions can be drawn from the analysis of the experimental data revealed by Figure 4:According to the values of the parameter pH_PZC_, the retention of the cationic dye MB occurs in the environment with a pH > 5, and the optimum is reached around the pH value = 9. The degree of recovery (R%) follows the same evolution curve as the maximum biosorption capacity (q) (Figure 4a,b). It is also observed that although the curves q = f (pH) and R (%) = f (pH) have the same allure in both types of biomass immobilization, the maximum values for q and R are different, respectively higher (104.45 mg/g, 29.09%) in the case of immobilization by microencapsulation with the help of the Buchi device compared to those obtained by a simple dropping technique (76.89 mg/g, 24.83%).Figure 4c,d show a decrease in the amount of dye retained per unit mass of biosorbent from 15.5 mg/g to 6.1 mg/g (Figure 4c) and respectively from 50.803 mg/g to 6.537 mg/g (Figure 4d) as the amount of biosorbent increases from 0.264 g/L to 3.08 g/L. There is also an increase in the percentage of dye recovered with the increase of biosorbent, which can be explained by the increase in the number of active positions favorable to biosorption as the amount of biosorbent increases. By analyzing the variations of the two parameters (q and R), it was established that a dose of 0.264 g d.w./L (respectively, 5.28 g/L biosorbent) in the case of the biosorbent obtained by a simple dripping technique, and 0.15 g d.w./L (respectively, 3 g/L biosorbent) in the case of biosorbent obtained by encapsulation using a Buchi microencapsulator can be considered optimal for the removal of MB dye from aqueous solutions.Figure 4e shows an increase in biosorption capacity with contact time, which increases faster in the first 100 min regardless of the diameter of the biosorbent particles, followed by a slower increase until equilibrium is reached.Figure 4g,f show an increase in biosorption capacity as the initial concentration of the dye in the aqueous solution increases until the saturation value of the biosorbent is reached. It also can be observed the positive influence of temperature increase on the biosorption capacity; in both cases, the biosorption process proving to be endothermic.

The conclusion from analyzing Figure 4 is that the biosorption of the cationic dye MB is much more favorable using a biosorbent with granules obtained by encapsulating the microbial biomass in the sodium alginate matrix. This is because the smaller particles provide a much larger contact surface between the dye and the microbial biomass, which favors biosorption.

### 3.4. Evaluation the Equilibrium of the Biosorption Process of Cationic Dye Methylene Blue onto Biosorbent by Immobilization of Microbial Residual Biomass onto Alginate Matrix

Starting from the obtained biosorption isotherms presented in Figure 4f,g and from the graphical representation of the linearized forms of the models of adsorption isotherms used (Freundlich, Langmuir, and Dubinin–Radushkevich) (Figure 5), the characteristic quantitative parameters were calculated and systematized in Table 2.

Studying the data presented in Table 2, the following considerations may be issued: ➢The values of the calculated quantitative parameters are a function of temperature and have higher values in the case of smaller diameter granules (obtained by encapsulated using a Buchi microencapsulator) because in this case, a larger specific contract area is ensured, which facilitates the biosorption process.➢Temperature dependence suggested an endothermic biosorption process favored by relative high temperature.➢Taking as appreciation criterion the values of the regression coefficient R^2^ and the values of the maximum biosorption capacities (q_0_, mg/g), among the two linearized forms of the Langmuir isotherm, the form I is best suited to model the data in the case of biosorbent obtained by a simple dropping technique, and the form II in the case of biosorbent obtained by encapsulation using a Buchi microencapsulator.➢The mean free biosorption energy, E, calculated by DR equation, can be useful to estimate the nature of the biosorption process (physical or chemical) [33]. In this case, there are two distinct situations: in the case of biosorbent obtained by a simple dripping technique the energy values, E, are between 13.87–15.4 kJ/mol suggesting for the process of MB dye biosorption an ion exchange mechanism (the sorption energy is within 8–16 kJ/mol). In the case of the biosorbent obtained by encapsulation using a Buchi microencapsulator, the energy values, E, are lower, in the range 7.4–7.9 kJ/mol for temperature between 5–17 °C and 10 kJ/mol for 45 °C, which suggests a process of biosorption of the same MB dye by physical mechanism as a result of the electrostatic interaction bonds, for the lower temperatures and ion exchange for the higher temperature. This behavior of biosorbents differs not only from what we found in the case of our studies about biosorption the reactive dyes [14,16] but also between these two types of biosorbents and can be explained by the structure and size of the dye molecule. Methylene blue dye has a much smaller molecule than the previously studied dye (Brilliant Red HE-3B reactive dye with MW 1463), which implies a higher probability to penetrate the internal structure of the biosorbent. Thus, its retention can be explained both on the basis of the formation of chemical and physical bonds (hydrogen bonds, van der Waals, dipole–dipole, etc.) made with the functional groups existing in the biosorbent structure and on its surface. On the other hand, in the case of smaller diameter biosorbent beads, the bisorption process is determined by the physical binding of the dye molecules only on the biosorbent surface due to a larger granule–dye contact surface and the more compact structure of the granules. This behavior could also explain why the values of the maximum biosorption capacities, q_0_ (represents the total specific meso- and macropore volume of the biosorbent, mg/g), calculated with the DR model have values closer to those resulting from the Langmuir model in the case of biosorbent obtained by a simple dropping technique with granules diameter size φ 1 = 4 mm.

The values obtained for the biosorption capacity, according to the Langmuir I and II model for both types of microbial biosorbent at room temperature, are comparable with other biosorption capacities, reported in the literature, for other types of microbial biomass immobilized on various supports for removal of methylene blue from aqueous medium (Table 3).

Depending on the immobilization technique, different biosorption capacities were obtained, respectively, between 47.62–87.72 mg/g in the case of biosorbent obtained by a simple dropping technique, and 188.68–434.78 mg/g in the case of biosorbent obtained by encapsulation using a Buchi microencapsulator (Table 2). These values were compared with the data in the literature (Table 3), and it is observed that this new biosorbent based on residual microbial biomass of *Saccharomyces pastorianus* immobilized in sodium alginate matrix led to appreciable values, comparable to what has been achieved previously. This fact demonstrates that the immobilization of residual microbial biomass in the sodium alginate matrix could be a form of recovery of this manufacturing waste.

## 4. Conclusions

The results on methylene blue dye biosorption on biosorbents based on residual microbial biomass of *Saccharomyces pastorianus* obtained by two immobilization techniques (a simple dropping technique and encapsulated using a Buchi microencapsulator) confirms that the process depends on the pH of the dye solution, the amount of dry matter in the biosorbent, temperature, and phase contact.

The biosorption process is based on a physical or chemical mechanism, depending on the size of the biosorbent granules, as suggested by the energy values of the process, E, obtained using the DR model: an energy values, E, between 13.87–15.4 kJ/mol (in the case of bigger particles of biosorbent obtained by a simple dropping technique) suggests for the process of biosorption of the MB dye an ion exchange mechanism. In the same time, a lower energy E, in the range 7.4–7.9 kJ/mol (in the case of smaller particles of biosorbent obtained by encapsulated using a Buchi microencapsulator) suggests a process based on physical mechanism as a result of the electrostatic interaction bonds.

Modelling based on the Langmuir isotherm allowed the calculation of the maximum biosorption capacity, which is between 47.62–87.72 mg/g in the case of biosorbent obtained by a simple dropping technique, and 188.68–434.78 mg/g in the case of biosorbent obtained by encapsulation using a Buchi microencapsulator, values depending on the size of the biosorbent granules.

These results confirm that the residual microbial biomass immobilized in a polymeric matrix such as sodium alginate can be considered a biomaterial with efficient biosorbent properties and in retaining cationic organic dyes present in aqueous solutions in moderate concentrations.

## Figures and Tables

**Figure 1 polymers-14-02728-f001:**
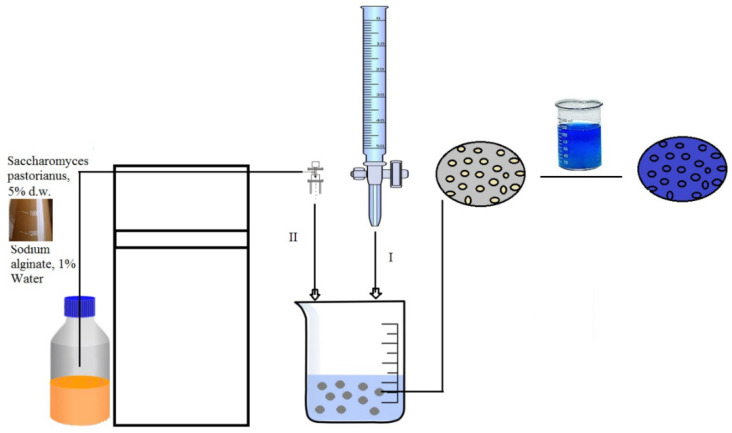
Systematization of procedures for obtaining biosorbents based on residual microbial biomass immobilized in sodium alginate matrix: (I) a simple dripping technique; (II) encapsulated using a Buchi microencapsulator.

**Figure 2 polymers-14-02728-f002:**
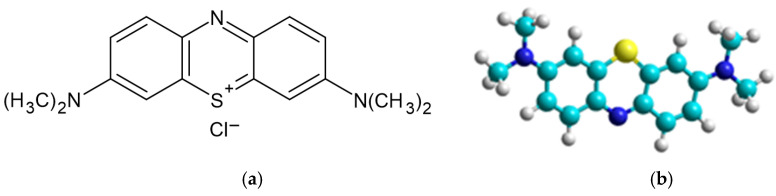
Chemical (**a**) and molecular (**b**) structure of cationic dye, methylene blue.

**Figure 3 polymers-14-02728-f003:**
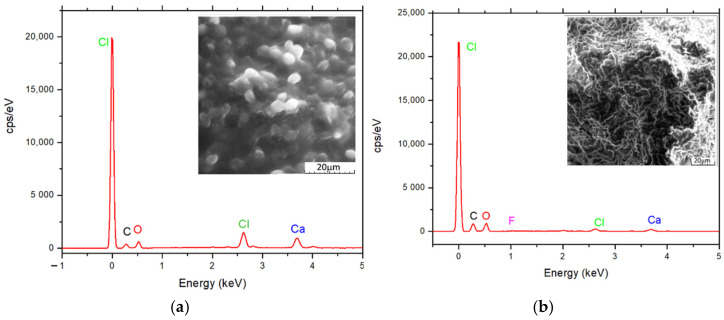
Scanning electron microscopy (SEM) and EDAX spectrum of the obtained polymeric microbial biosorbent based on *Saccharomyces pastorianus*, immobilized using the two techniques: (**a**) a simple dropping technique; (**b**) encapsulated using a Buchi microencapsulator.

**Figure 4 polymers-14-02728-f004:**
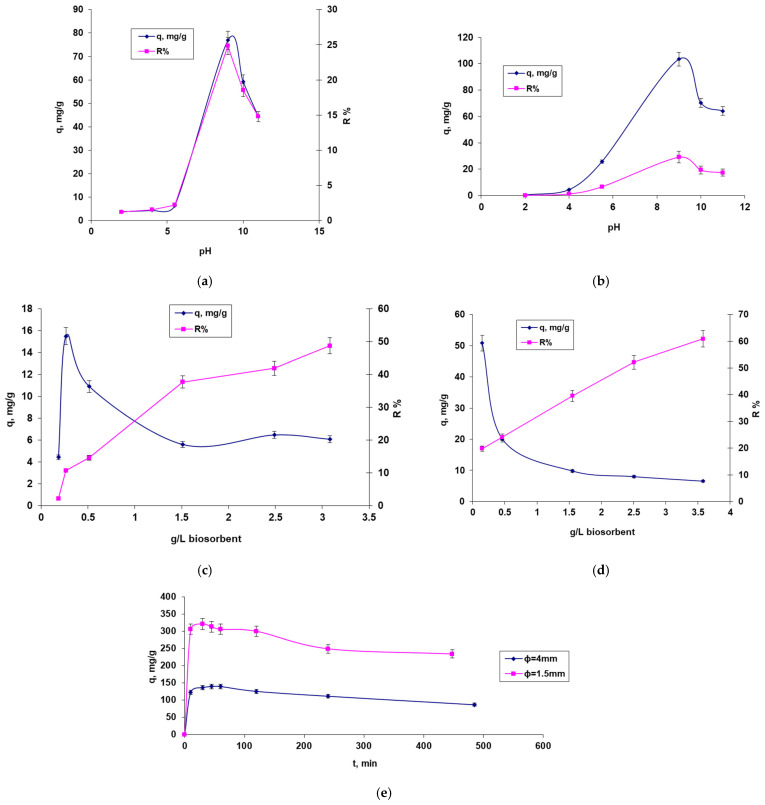
Factors influencing the biosorption of the cationic dye methylene blue dye onto microbial biosorbent obtained by a simple dripping technique (**a**,**c**,**f**) and by encapsulation using a Buchi microencapsulator (**b**,**d**,**g**); (**a**,**b**) the influence of pH and biosorbent production technique; (**c**,**d**) the influence of biosorbent dose and biosorbent production technique; (**e**) the influence of contact time between phases; (**f**,**g**) the influence of temperature and initial concentration of the dye solution.

**Figure 5 polymers-14-02728-f005:**
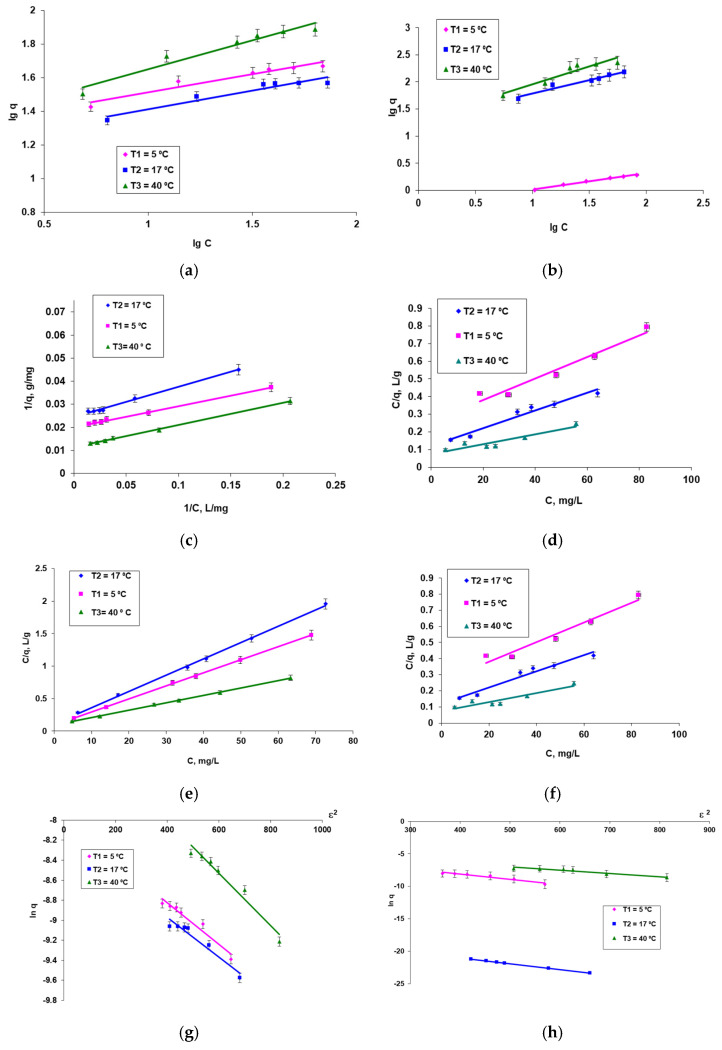
Linearized form of Freundlich (**a**,**b**), Langmuir I (**c**,**d**), Langmuir II (**e**,**f**), and DR (**g**,**h**) plots for the methylene blue cationic dye on biosorbent based on *Saccharomyces pastorianus* residual biomass obtained by a simple dripping technique (**a**,**c**,**e**,**g**) and by encapsulation using a Buchi microencapsulator (**b**,**d**,**f**,**h**) in sodium alginate. Conditions: pH = 3, contact time = 24 h, biosorbent amount: 0.26 g/L (with 5%d.w. for φ1 = 4 mm), and 0.15 g/L (with 5% d.w. for φ2 = 1.5 mm); diameter of granules: φ 1 = 4 mm (**a**,**c**,**e**,**g**) and φ 2 = 1.5 mm (**b**,**d**,**f**,**h**) at three temperatures: 5°, 17°, and 40 °C.

**Table 1 polymers-14-02728-t001:** Physical–chemical parameters that influence the biosorption of methylene blue dye onto microbial biosorbent.

Parameters	Studied Limits of Variation
pH	2–11
T, °C	5, 20, 60
t, min	10 min–24 h
Biosorbent dose, g/L	2.4–50.4
Initial dye concentration in solution, mg/L	14.32–229.2
Granule size: ➢ obtained by a simple dripping technique ➢ encapsulated using a Buchi microencapsulator	φ = 4 mm φ = 1.5 mm

**Table 2 polymers-14-02728-t002:** Characteristic parameters for the biosorption of methylene blue cationic dye onto biosorbent based on residual biomass of *Saccharomyces pastorianus* immobilized in alginate matrix.

Isotherm	φ 1 = 4 mm			φ 2 = 1.5 mm		
	278 K	290 K	313 K	278 K	290 K	313 K
Freundlich						
K_F_ ((mg/g)(L/mg)^1/n^)	15.59	19.87	8.79	0.51	19.40	19.59
n	4.52	4.64	2.89	3.29	2.01	1.50
R^2^	0.91	0.93	0.94	0.99	0.95	0.90
Langmuir						
Langmuir I						
q_0_ (mg/g)	49.75	40.81	88.49	204.08	188.67	434.78
K_L_ (L/g)	0.22	0.18	0.11	0.01	0.04	0.02
R^2^	0.99	0.99	0.99	0.96	0.97	0.97
Langmuir II						
q_0_ (mg/g)	47.61	40.00	87.71	163.93	200.00	227.27
K_L_ (L/g)	0.22	0.22	0.12	0.02	0.04	0.01
R^2^	0.99	0.99	0.99	0.95	0.95	0.71
Dubinin-Radushkevich (DR)						
q_0_ (mg/g)	108.37	90.18	301.74	2364.63	8.413 × 10^−3^	3527.96
β (mol^2^/kJ^2^)	0.002	0.002	0.002	0.008	0.0089	0.005
E (kJ/mol)	15.43	15.81	13.867	7.906	7.495	10.00
R^2^	0.9400	0.9300	0.9598	0.9305	0.9991	0.9159

**Table 3 polymers-14-02728-t003:** Applications of various microbial biomass-based biosorbents immobilized form for methylene blue dye removal.

Biosorbent	Conditions	Maximum Adsorption Capacity, mg/g	References
Sodium alginates beads	pH = 9	0.25	[34]
*Lactarius piperatus*	T = 25 °C; pH = 7	384.6 ± 3.4	[18]
*Microspora* sp	pH = 7, 150 rpm, 24 h	139.11	[19]
*Chlamydomonas moewusii*	pH 10, 7 h	212.41	[20]
*Sargassum ilicifolium*	0.6 h	99.7	[29]
*Brewer’s spent grain*	7 h	298.35	[30]
*Sargassum hemiphyllum*	2 h, pH = 5	729.93	[31]
*Trichoderma viride, entrapped in loofa sponge*	90 min, pH = 10	201.52	[35]
*Bacillus subtilis immobilized in calcium alginate*	30 °C, 20 g/L biosorbent, shaking speed 900 rpm	90% removal	[36]
Residual biomass of *Saccharomyces* pastorianus immobilized in sodium alginate by a simple dropping technique	pH = 3; t = 24 h; amount of biosorbent = 0.26 g/L (with 5% d.w); φ = 4 mm; T = 5–40 °C	47.62–87.72	This study
Residual biomass of *Saccharomyces* *pastorianus* immobilized in sodium alginate by encapsulation using a microencapsulator	pH = 3; t = 24 h; amount of biosorbent = 0.15 g/L (with 5% d.w); φ = 1.5 mm; T = 5–40 °C	188.68–434.78	This study

## Data Availability

Not applicable.

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
