# Peer review of "Biosorbents Based on Biopolymers from Natural Sources and Food Waste to Retain the Methylene Blue Dye from the Aqueous Medium"

_polymers, 2022, doi:10.3390/polym14132728_

Round 1

Reviewer 1 Report

In this article, a biosorbent based on residual biomass from brewing industry (Saccharomyces pastorianus) immobilized in a natural biopolymer (sodium alginate) was investigated for Methylene Blue removal from aqueous medium.

The specific comments are as follows:

1. The right side of Chemical structure of cationic dye (Figure 2. (a)) is incomplete.

2. Some counterparts articles should be referred, such as Chemical Engineering Journal, 428, 131370 about selective adsorption of anionic dyes. This work might provide some useful information to further understand your work.

3. The English writing of this article does not meet the standards of this journal, and the full text needs to be carefully revised.

Reviewer 2 Report

The authors report the study of a material based on alginate and Saccharomyces pastorianus as a biosorbent for methylene blue dye from aqueous solutions. The conditions for optimal sorption capacity are investigated.

The manuscript is well structured and comprehensible. However, in addition to some necessary language improvements, there are some issues to be addressed: 

The authors should provide more details regarding the synthetic methods. A figure and a couple of references are not helpful enough, more information needs to be added in the text.

The authors mention that the granules are stored in 2% CaCl2 until use. Is that the concentration of the gelation bath? For how long are the granules stored before use? Can this time affect the sorption capacity?

(Also, the claim that the reason for calcium and chloride detection by EDAX is that the granules are stored in CaCl2 solution is inaccurate; Ca2+ ions have replaced Na+ ions of sodium alginate for the gelation to occur, thus calcium is actually part of the material; chloride, on the other hand, is present in most alginate samples.)

The authors mention the importance of high porosity and surface area for sorption applications but don't report relevant values for their materials. Were porosities and surface areas measured?

The authors determine the dye removal from the aqueous solution by optical spectroscopy. Given that methylene blue has a high abrosptivity, are the authors certain that absorbance values are not too high to be raliably related to dye concentrations? Addition of spectra or absorbance values to the manuscript is advisable.

Ca2+-gelated alginate could adsorb methylene blue without the incorporation of a yeast. Do the authors have any data (either from their own experiments or from citing literature) on the sorption capacity of alginate as the single sorbent and how it compares to that of their material? Other than the sorption capacity, can the authors clarify the advantages Saccharomyces pastorianus adds to a sorbent?

The optimum sorption conditions found by the authors are at pH = 9. Can the authors add a comment regarding how this pH value compares to usual pH values of actual wastewater samples?

Please check the legends of the graphs in Figure 4.

Has the recyclability of the reported sorbents been examined? If possible, addition of relevant experiments would improve the significance of the reported work.

The authors mention that the encapsulation method gives materials with better sorption capacity, and attribute this to their smaller size, which results to a higher contact surface with the dye. If particle size is the only advantage of the encapsulation method over the dripping method, it would be advisable that the authors tried to fabricate smaller beads by the dripping method (i.e., using a narrower tip) and compare results to those already reported.

It would be helpful to present representative curves of the sorption equilibrium models used.

Reviewer 3 Report

The present manuscript studies the capacity of removal of methylene blue by biomass of Saccharomyces pastorianus immobilized with two procedures. The manuscript is a typical biosorption assay. The results are interesting but the manuscript is of poor quality. I consider that it should be clearly improved and fix some problems before publication. In its current state, this manuscript should be rejected. If the authors modify it appropriately, this decision could be reconsidered.

1) The Title is too general. The Title should include the material that has been used, and that this material was immobilized. I consider that the Title is not very representative of the content of the manuscript.

2) Biosorption experiments: how many replicates were used to perform these experiments? It is not appropriate to do a single experiment. What reproducibility can be deduced? What variation could have the obtained values?. Since the deviations at each point are not shown in the figures, it can be assumed that the experiment was not replicated. This is a serious methodological error that should prevent publication of the manuscript. explain.

3) Lines 224-228: To avoid this, the two figures should have the same y-axis scale (120 for q, and 35 for %). It is not easy (visually) to compare the results of both materials if the scale of the figures is different.

4) Lines 255-267: these are not results, this is Material and Methods.

5) Table 2: Although the equations of isotherms are well known, the way of putting these equations in Table 2 is not very elegant, and the meaning is not very clear. The equations should be better explained in a Material and Methods section or more developed in Supplementary Material.

6) Explain in Material and methods how the adjustment to the equations was made. In general, in this manuscript, Material and Methods is very poor.

7) Table 2: Are so many decimal places really necessary?

8) Line 293: I would not consider ion exchange to be a chemical mechanism.

9) Line 301: What is the dye previously studied?.

10) Lines 303-304: But then the efficiency will be lower. This explanation is a bit confusing considering the results.

11) Taking into account the results, with the consequent controversy, the authors should propose a biosorption model of this dye to this material.

12) Table 3 is not suitable for this manuscript. Make a table considering only methylene blue. This comparison makes no sense, since the affinity for other dyes can be very different.

13) Taking into account my previous comment, this manuscript lacks a Discussion section. This is an aspect that clearly needs to be improved. For example, compare this material to other materials that have been used to remove this dye. What advantages does this biomass have over others? was it more efficient?

14) The Bibliography does not have a uniform format.

15) In the Bibliography, the names of the species must be written in italics.

Round 2

Reviewer 2 Report

The author's have sufficiently answered the reviewer's comments. The only minor issue remaining is probably due to lack of clarity on the reviewer's part; the comment about the legends in figure 4 referred to the inset legends in graphs (f) and (g). In (f), the blue line corresponds to 5 oC and the pink line to 17 oC, while in (g) it's the other way round. The author's should check if this is right. Otherwise the manuscript can be accepted for publication.

Reviewer 3 Report

The modifications in the manuscript have been correct and the answers to the questions have been adequate. The manuscript could be published, but some minor corrections are necessary:

1) If the experiments were performed in triplicate, the figures must have the deviations. This is important to continue publication of the manuscript.

2) The equations must be numbered. In general this section should be better ordered.

3) Line 218: What does "characteristic sizes were determined" mean?

4) Line 289: too many decimal places.

5) Add in Table 3 the results of this work (qmax).
